# OpenReview forum: "A Frustratingly Simple Yet Highly Effective Attack Baseline: Over 90% Success Rate Against the Strong Black-box Models of GPT-4.5/4o/o1"
_NeurIPS.cc/2025/Conference — NeurIPS 2025 poster_

### Official Review · Reviewer_ZL92 · 2025-06-25

**Clarity:** 3
**Significance:** 3
**Originality:** 2
**Rating:** 2
**Confidence:** 4

**Summary:**

This paper investigates transferable adversarial attacks under the standard black-box threat model. It introduces a novel method, M-Attack, which leverages input transformations, such as random image cropping, to extract local views of the input. This approach is motivated by the observation that previous methods fail to transfer effectively to recent commercial models. The proposed method demonstrates strong performance against commercial large vision-language models (LVLMs), including GPT-4.5, GPT-4o, Gemini, and Claude-3.7.

**Questions:**

- Does a non-uniform perturbation distribution correlate with stronger transferability? If so, can the authors provide empirical evidence that their method improves this property, perhaps using the ECDF analyses shown in Figures 3 and 4?

- Could the authors report standard evaluation metrics on open-source models to enable easier comparison with prior work? For example: Zero-shot classification with CLIP, Image-text retrieval tasks, Captioning metrics such as BLEU and CIDEr with large VLMs. These evaluations should ideally be conducted on widely used datasets such as ImageNet, MSCOCO, OK-VQA, etc. Given that the proposed method fundamentally targets CLIP encoders, this would be a relevant and fair comparison.

- This is more like a suggestion rather than a question. I believe the proposed method has the potential to serve as a generalizable enhancement to existing adversarial attacks rather than functioning solely as a standalone method. It would be particularly interesting to see how the input transformation technique could be integrated into or improve the performance of existing attacks mentioned in the strengths and weaknesses section.


- Could the authors elaborate on how KMRScore reduces human bias, given that the semantic keyword labels are still manually defined? A deeper discussion on potential limitations or mitigations of this subjectivity would be appreciated.


Addressing the first three points raised in my review is likely to change my current opinion on the rating.

Overall, I believe this work has strong potential and introduces valuable insights. However, the limitations in its current presentation, particularly the lack of thorough discussion of related work, insufficient evaluation of standard benchmarks, and limited analysis supporting its core observations. In my opinion, this currently outweighs its strengths.

**Ethical Concerns:**

["NO or VERY MINOR ethics concerns only"]

**Final Justification:**

Overall, I believe this work has strong potential and introduces valuable insights. The rebuttal successfully addresses some of my earlier concerns, and I find it particularly exciting that the proposed input transformation technique enhances the performance of existing baselines, suggesting it could serve as a broadly applicable and effective strategy in this domain.

However, since the method relies on CLIP encoders as surrogate models, I believe it is essential to include comparisons against existing CLIP-based attacks using standard evaluation protocols, as well as evaluations in VLM settings. These results were not provided in the rebuttal and would likely require substantial additions or changes to the current manuscript. Therefore, in my view, the paper is not yet ready for acceptance.

**Limitations:**

Yes

**Quality:**

2

**Strengths And Weaknesses:**

**Strengths**

- The paper identifies a consistent failure pattern in prior attacks, characterized by vague or ambiguous model outputs, an observation well-supported by experiments across multiple baselines.
- It offers a novel and insightful explanation for these failures: the uniform-like distribution of perturbations, which limits semantic specificity and transferability.
- The proposed solution generating semantically meaningful adversarial perturbations aligns with trends in recent and concurrent research, suggesting a promising direction for future work.
- Extensive experiments on commercial black-box LVLMs (e.g., GPT-4.5, GPT-4o, Gemini, Claude) demonstrate the practical effectiveness of the method in realistic threat scenarios.

**Weaknesses**

- While the observation in Section 3 regarding uniform-like perturbation distributions is insightful, the paper lacks direct evidence that the proposed method effectively addresses this issue.
- The novelty of the proposed method is somewhat limited. Input transformation techniques, particularly image cropping, are well-established in the literature as effective tools for improving transferability [1]. However, the authors did not discuss their method to prior works in this area.
- Although evaluations on real-world commercial black-box models are valuable, the paper would benefit from additional experiments under standard benchmark settings using open-source models to ensure fair comparisons and reproducibility.
- The authors claim that KMRScore reduces human bias, but the manual annotation of multiple semantic keywords per image introduces subjectivity and may not effectively mitigate bias. Similar concerns apply to the use of LLM-as-a-judge for evaluation.
- The proposed method is essentially a CLIP-based adversarial attack, using CLIP encoders as surrogates. Given that many large VLMs are built on CLIP, the effectiveness of the attack is perhaps unsurprising. The paper should explicitly discuss this aspect and include comparisons with other CLIP-based attack baselines. Several relevant works along the line of CLIP-based adversarial attacks are not mentioned in the related work section. I have included a few representative works [2] in this line of research, though the list is not exhaustive.



[1]  Xie, C., Zhang, Z., Zhou, Y., Bai, S., Wang, J., Ren, Z., & Yuille, A. L. Improving transferability of adversarial examples with input diversity. In CVPR, 2019\
[2] Zhang, J., Yi, Q., & Sang, J. Towards adversarial attack on vision-language pre-training models. In ACM MM, 2022\
[3] Lu, D., Wang, Z., Wang, T., Guan, W., Gao, H., & Zheng, F.  Set-level guidance attack: Boosting adversarial transferability of vision-language pre-training models. In ICCV, 2023 \
[4] He, B., Jia, X., Liang, S., Lou, T., Liu, Y., & Cao, X. (2023). Sa-attack: Improving adversarial transferability of vision-language pre-training models via self-augmentation. arXiv preprint arXiv:2312.04913. \
[5] Zhang, P. F., Huang, Z., & Bai, G. Universal adversarial perturbations for vision-language pre-trained models. In SIGIR, 2024 \
[6] Fang, H., Kong, J., Yu, W., Chen, B., Li, J., Wu, H., ... & Xu, K. (2024). One perturbation is enough: On generating universal adversarial perturbations against vision-language pre-training models. arXiv preprint arXiv:2406.05491.

---

Miscellaneous (not part of the rating justification):

There is a recent concurrent preprint (after NeurIPS deadline) that discusses ensemble surrogate models for attacking CLIP, which may be relevant to this work and could offer some insights into the ensemble strategy employed in M-Attack.

Huang, H., Erfani, S., Li, Y., Ma, X., & Bailey, J. (2025). X-Transfer Attacks: Towards Super Transferable Adversarial Attacks on CLIP. arXiv preprint arXiv:2505.05528.

---

> ### Author Rebuttal · Authors · 2025-07-31
>
> We thank the reviewer for the constructive and valuable feedback, which is greatly appreciated and will definitely help us improve the quality of our paper. We will accommodate all suggestions into the revised manuscript. Below, we provide detailed responses to each of the reviewer's comments.
>
> >**W1**: Regarding uniform-like perturbation distributions is insightful, direct evidence for the proposed method.
>
> Thanks for the constructive suggestion. We provide the numerical comparisons of ECDF distance to the uniform distribution in the table below with and without our local crop approach as the direct evidence, we calculate the KL divergence from a distribution to the uniform distribution as follows:
>
> Let the empirical distribution be defined as:
> $P(v) = \frac{n_v}{N}$
> where $n_v$ is the count of value $v$, $N$ is the total number of samples.
> Let the uniform probability mass function (PMF) be $U(v) = \frac{1}{K}$. Then, the KL divergence from P to U is given by:
> $D_{\mathrm{KL}}(P \parallel U_K)
> = \sum_{v=-16}^{16} \frac{n_v}{N} \log\left( \frac{n_v}{N} \cdot K \right)
> = \frac{1}{N} \sum_{v=-16}^{16} n_v \log\left( \frac{n_v K}{N} \right)$.
> As the perturbations are discrete, there are 33 possible perturbation values ranging from -16 to +16.
>
> Our Results Summary:
>
> |Setting	| KL Divergence	|Acc. (GPT-4o, Gemini-2.0, Claude-3.5)|
> |:-|:-:|:-:|
> Baseline (failed samples)|	0.012|	—
> Ours (w/o local crop)	|0.014	|(0.10, 0.08, 0.08)
> Ours (with local crop)	|**0.038**	|**(0.95, 0.78, 0.29)**
>
> These results show that our method increases KL divergence from the uniform distribution, indicating more non-uniform perturbations. The corresponding boost in attack success rate further confirms the effectiveness of our strategy. Detailed values are provided in **Q1**'s response.
>
> >**W2**: Input transformation techniques, particularly image cropping, are well-established for improving transferability [1]. Discuss their method to prior works in this area.
>
> Thanks for sharing the relevant paper [1]. To clarify, [1] focuses on generating adversarial examples by increasing input diversity, i.e., applying random transformations to the input images at each iteration to enhance robustness and effectiveness. In contrast, our method does **not aim to increase input diversity**, but instead focuses on discovering a more effective **semantic matching strategy between the source and target images**. While there may be surface-level similarities in implementation (e.g., input manipulation or crop across iterations), the underlying motivation and high-level objectives are fundamentally different.
>
> Moreover, our approach is specifically designed for and evaluated against the strongest black-box LVLMs to date, where we demonstrate state-of-the-art attack performance across GPT-4o, Gemini, and Claude models. As suggested, we will include a detailed discussion in the revised manuscript to further clarify the differing assumptions and goals.
>
> >**W3 & Q2**: **W3**) The paper would benefit from additional experiments under standard benchmark settings using open-source models. **Q2**) Could the authors report standard evaluation metrics on open-source models?
>
> Yes, following the suggestion, we provide more results on standard benchmark settings (image captioning using ImageNet as source and COCO as target, and VQA on OK-VQA) using open-source models in the tables below:
>
> **Image Captioning**:
>
> *Source dataset: ImageNet & Target dataset: COCO2014 Val*.
>
> **Reference paper**:
>
> AnyAttack: Towards Large-scale Self-supervised Adversarial Attacks on Vision-language Models. In CVPR 2025.
>
> **Results:**
>
> | **AnyAttack\|M-Attack (Ours)**      | **SPICE** | **BLEU-1** | **BLEU-4** |**METEOR** |**ROUGE-L** |**CIDEr** |
> |:-|:-:|:-:|:-:|:-:|:-:|:-:|
> | BLIP       | 4.13\|**12.02** | 46.32 \|**65.71** | 3.13\|**23.12** | 11.38 \|**21.07** | 33.32 \|**46.82** | 18.28\|**86.23** |
> | BLIP2     | 4.48\|**8.69** | 46.68\|**53.29** | 5.96\|**13.52** | 11.38\|**15.43** | 33.44\|**38.52** | 20.20\|**44.25** |
> | InstructBLIP | 5.89\|**15.14** | 38.79\|**51.76** | 3.83\|**11.57** | 12.77\|**20.91** | 28.36\| **39.55** | 16.63\| **43.47** |
>
> **Visual Question Answering (OK-VQA Dataset)**:
>
> **Reference paper:**
>
> On the Robustness of Large Multimodal Models Against Image Adversarial Attacks. In CVPR 2024.
>
> **Results:** (*Pre* refers to accuracy for pre-attack.)
>
> | **VQA Accuracy (%)**  | **BLIP2 $\downarrow$** | **InstructBLIP $\downarrow$** | **LLaVA1.5 $\downarrow$** |
> |:-|:-:| :-: | :-: |
> | Pre       | 25.00 | 33.60 | 30.20 |
> | AnyAttack | 7.20 | 21.80 | 24.80 |
> | M-Attack (Ours) | **6.80** | **13.40** | **12.40** |
>
> >**W4 & Q4**: **W4**) The authors claim that KMRScore reduces human bias, but the manual annotation of multiple semantic keywords per image introduces subjectivity and may not effectively mitigate bias. **Q4**) Could the authors elaborate on how KMRScore reduces human bias? A deeper discussion.
>
> Thank you for highlighting this important concern. We clarify our rationale and the steps taken to mitigate bias as follows: 1)	While the semantic keywords are manually defined, we follow task-specific, consistent guidelines. Each keyword corresponds to an objectively verifiable element (e.g., object count, color, spatial relation) from the reference answer. Once defined, the same keyword set is applied uniformly across all model outputs to reduce per-sample subjectivity. We also release the full keyword sets and annotation criteria for transparency and reproducibility. 2) KMRScore provides a structured alternative to free-form human judgment by decomposing evaluation into discrete, checkable semantic units. This improves interpretability and reduces susceptibility to stylistic or linguistic bias. 3) LLM-as-a-Judge Limitations: We agree that LLM-based evaluation introduces bias from pretraining. In our setting, KMRScore shows greater consistency and better alignment with ground-truth semantics, especially important for adversarial success evaluation, where accurate concept transfer is more critical than fluency.
>
> We will expand the manuscript to further discuss these trade-offs and emphasize that our goal is to reduce (not eliminate) bias through structured and reproducible evaluation.
>
> >**W5**: The proposed method is essentially a CLIP-based adversarial attack, using CLIP encoders as surrogates. Explicitly discuss this aspect and include comparisons with other CLIP-based attack baselines.
>
> Thanks for the helpful feedback and for pointing out these relevant works. We agree that our method leverages CLIP as a surrogate, and this is indeed a key design choice. However, we emphasize that **the strength of our approach lies in its ability to transfer adversarial semantics effectively to strong, unseen black-box LVLMs** (e.g., GPT-4o, Claude-3.5), which are not directly optimized via CLIP gradients. The success is not solely due to shared CLIP foundations but also stems from our carefully designed local-global matching and semantic reinforcement strategies.
>
> We appreciate the suggested references. We will revise the related work section to incorporate and discuss these CLIP-based attacks more thoroughly. We will also include comparisons with these representative baselines to better contextualize our contributions.
>
> >**Q1**: Does a non-uniform perturbation distribution correlate with stronger transferability? If so, can the authors provide empirical evidence, perhaps using the ECDF analyses shown in Figures 3 and 4?
>
> Yes, following the suggestion, we provide the detailed ECDF results before and after applying our method, as shown below:
>
> Reference (20 failed samples) deviation from uniform distribution (%):
> [-1.23, -2.12, -1.71, -0.57, -1.56, -0.65, -2.02, -1.13, -1.48, -1.14, -1.28, -1.42, -0.77, -1.25, -1.08, +2.06, +6.53, +0.87, +1.00, +0.59,  +1.56, +0.89, +1.72, +1.67, +0.73, +0.68, +0.31, +0.28, -0.74, +0.32, +0.02, -0.33, -1.23].
>
> Baseline (without crop deviation from uniform distribution (%):
> [+0.88, -0.66, +4.86, -0.50, +0.22, +3.16, -1.82, -1.18, +0.38, -0.66, -2.39, -0.41, -0.72, +1.21, -3.33, -1.40, +2.30, +3.71, -3.02, +1.29, -1.96, -1.71, -1.74, -0.85, +1.99, +1.42, +0.12, -0.73, +0.01, -0.21, +1.59, -1.26, -0.01].
> **Acc: (0.1, 0.08, 0.08)**  for GPT-4o, Gemini-2.0, Claude-3.5.
>
> Ours (with crop, deviation from uniform distribution (%):
> [+11.77, +2.41, -1.77, -8.44, -8.51, -13.62, -9.38, -13.59, -5.62, -9.64, +0.89, -4.03, +7.72, +1.84, +14.58, +13.00, +78.59, +5.74, +11.17,  -1.99, +3.93, -7.47, -2.95, -13.06, -8.57, -15.40, -10.80, -14.73, -9.05, -8.83, -1.62, +3.59, +13.82].
> **Acc: (0.95, 0.78, 0.29)** for GPT-4o, Gemini-2.0, Claude-3.5.
>
> Across different target models (GPT-4o, Gemini-2.0, Claude-3.5), our method consistently demonstrates better transferability in terms of both accuracy and distributional deviation.
>
> >**Q3**: This is more like a suggestion rather than a question. It would be particularly interesting to see how the input transformation technique could be integrated into or improve the performance of existing attacks.
>
> Thank you for the kind suggestion. Following the reviewer's advice, we have applied our method to the AnyAttack approach, and the results are presented in the table below.
>
> |    **Method**      | **GPT‑4o&nbsp;(ASR,&nbsp; a / b / c)** | **Gemini-2.0&nbsp;(ASR,&nbsp; a / b / c)** | **Claude-3.5&nbsp;(ASR,&nbsp; a / b / c)** |
> |:--|:--:|:--:|:--:|
> | AnyAttack       |0.42,  0.44 / 0.20 / 0.04| 0.48, 0.46  / 0.21 / 0.05 | 0.23, 0.25 / 0.13 / 0.01 |
> | AnyAttack w/ our crop matching | **0.54, 0.46 / 0.23 / 0.05** | **0.50, 0.48 / 0.23 / 0.06** | **0.30, 0.29 / 0.21 / 0.03** |
>
> >Overall, I believe this work has strong potential and introduces valuable insights. However, the limitations in its current presentation.
>
> We appreciate the kind comments and we are encouraged by the recognition of our work's potential, we will revise our paper carefully based on the reviewer's feedback and suggestions.

---

> > ### Comment · Reviewer_ZL92 · 2025-08-05
> >
> > Thank you to the authors for their thoughtful responses. While some of my questions have been addressed, I remain unconvinced about the technical novelty of the proposed method. The core idea, applying random transformations or augmentations, is a well-established technique known to enhance transferability in this domain.
> >
> > Moreover, the proposed method leverages CLIP as a surrogate, which aligns with an existing line of work that has demonstrated similar effectiveness on black-box LVLMs. I recommend that the authors include a comparison with these related methods in the revision, as noted in my initial review. This should include evaluations on both standard open-source benchmarks (datasets and victim models) and the commercial closed-source victim models used in this paper.

---

> > > ### Author Response · Authors · 2025-08-07
> > > **Further Response to Reviewer ZL92 [1/2]**
> > >
> > > We sincerely appreciate the reviewer's follow-up comments. We understand the reviewer's concern that our method may appear simple using two different input transformations by cropping-based matching on the source and target images for optimization. However, we would like to highlight two key points:
> > > 1. The core idea of our method is not merely about applying input transformation or data augmentation, but about leveraging localized regions between source and target data for more effective alignment. The reviewer may have overlooked the experimental results we presented regarding different data augmentations. The key in our framework is to design the alignment, the final formulation we present is indeed simple and elegant, while we highlight that no prior work has achieved such significant performance gains on competitive black-box LVLMs. The simplicity of the final approach does not reflect the effort behind it: **we spent substantial time, experimentation, and analysis to identify and isolate the most effective components**. It would be unfair to dismiss the contribution based solely on the apparent simplicity of the final design. We deliberately chose to present the most impactful and concise version of the method. In fact, simplicity and effectiveness are qualities we aim for, and we also provide theoretical justification in the appendix explaining why our approach works.
> > >
> > > 2. There is strong precedent for simple yet highly effective methods making meaningful impact in the community. We view this as a strength, not a weakness. For instance:
> > >
> > >    • TFA [1] finds that simply fine-tuning the last layer of existing detectors on rare classes is key for few-shot object detection, outperforming complex meta-learning methods by 2–20%, and in some cases, doubling accuracy.
> > >
> > >    • PURE [2] proposes a simple pipelined approach for entity and relation extraction using two independent encoders. Despite its simplicity, it achieves state-of-the-art results on ACE04, ACE05, and SciERC.
> > >
> > > ---
> > > References:
> > >
> > > [1] Wang et al., Frustratingly Simple Few-Shot Object Detection, ICML 2020.
> > >
> > > [2] Zhong & Chen, A Frustratingly Easy Approach for Entity and Relation Extraction, NAACL 2021.
> > >
> > > Regarding the comment on "the lack of thorough discussion of related work", we have originally prepared an in-depth discussion for rebuttal, but due to space constraints, we have to compress it for the rebuttal. We provide them below and we have also clearly committed to expanding the related work section and discussing them in the revision. We feel that the absence of a longer related work discussion should not be a reason to reject, especially since the referenced works primarily target open-source models, while our focus is on the more challenging setting of closed-source black-box LVLMs. Furthermore, given that we have addressed nearly all of the reviewer's **experimental concerns with additional results—many of which required significant time and effort on our part**. Dismissing the paper despite this would be disproportionately unfair.
> > >
> > > Some more discussions with the reviewer mentioned related work are:
> > >
> > > M-DI2-FGSM [1] proposes to improve the transferability of adversarial examples by introducing input diversity, generating adversarial samples through random transformations of the input image at each iteration rather than using only the original image. Co-Attack [2] introduces a multimodal adversarial framework, Collaborative Multimodal Adversarial Attack, which simultaneously perturbs both the image and text modalities in vision-language pre-trained (VLP) models. SGA [3] proposes Set-level Guidance Attack, a highly transferable method that fully exploits cross-modal interactions and incorporates alignment-preserving augmentations guided by semantic consistency across modalities. Sa-attack [4] identifies two key factors affecting transfer attacks on VLP models: inter-modal interaction and data diversity. Based on this, they propose a self-augmentation-based method that applies distinct augmentations to images and text to enhance adversarial transferability. [5] introduces ETU, a black-box method for generating universal adversarial perturbations (UAPs) aimed at misleading a wide range of VLP models. ETU jointly considers UAP characteristics and cross-modal interactions, encouraging both global and local utility of perturbations across downstream tasks. C-PGC [6] is a contrastive-training perturbation generator conditioned on cross-modal inputs. It flips the objective of contrastive learning, typically used for alignment, by training on adversarial positive/negative image-text pairs to disrupt multimodal alignment. C-PGC also incorporates both unimodal and cross-modal cues for more effective attack guidance.

---

> ### Author Response · Authors · 2025-08-07
> **Further Response to Reviewer ZL92 [2/2]**
>
> ----
> References:
>
> [1] Xie, C., Zhang, Z., Zhou, Y., Bai, S., Wang, J., Ren, Z., & Yuille, A. L. Improving transferability of adversarial examples with input diversity. In CVPR, 2019.
>
> [2] Zhang, J., Yi, Q., & Sang, J. Towards adversarial attack on vision-language pre-training models. In ACM MM, 2022.
>
> [3] Lu, D., Wang, Z., Wang, T., Guan, W., Gao, H., & Zheng, F. Set-level guidance attack: Boosting adversarial transferability of vision-language pre-training models. In ICCV, 2023.
>
> [4] He, B., Jia, X., Liang, S., Lou, T., Liu, Y., & Cao, X. (2023). Sa-attack: Improving adversarial transferability of vision-language pre-training models via self-augmentation. arXiv preprint arXiv:2312.04913.
>
> [5] Zhang, P. F., Huang, Z., & Bai, G. Universal adversarial perturbations for vision-language pre-trained models. In SIGIR, 2024.
>
> [6] Fang, H., Kong, J., Yu, W., Chen, B., Li, J., Wu, H., ... & Xu, K. (2024). One perturbation is enough: On generating universal adversarial perturbations against vision-language pre-training models. arXiv preprint arXiv:2406.05491.
>
>
> Considering the limited time for further comments, we also provide the suggested comparisons with the related methods, our approach is still significantly better than them on the standard open-source benchmarks (various datasets and victim models):
>
> - ***Flickr30K test set**, $\epsilon=12$, surrogate model: CLIP-ViT-B/16*
>
> | **Target Model/Source Model**  |CLIP-ViT-B/16 | CLIP-ViT-B/16 | CLIP-ResNet50 |  CLIP-ResNet50 | ALBEF |  ALBEF | TCL |  TCL |
> |:-:|:-:|:-:|:-:|:-:|:-:|:-:|:-:|:-:|
> |  | **TR R@1/@5** | **IR R@1/@5** | **TR R@1/@5** | **IR R@1/@5** | **TR R@1/@5** | **IR R@1/@5** | **TR R@1/@5** | **IR R@1/@5** |
> | Co-Attack [1] | 93.25/84.88| 95.86/90.83 | 28.79/13.78 | 40.03/26.77| 23.60/1.87 | 36.48/11.74 |27.85/2.83|41.19/13.80|
> | SGA [2] | 99.08/97.25| 98.94/97.53 | 31.24/19.45 | 42.12/30.36| 33.38/2.46 | 44.16/13.21 |33.87/3.77|44.88/16.36|
> | ETU [3] | 93.13/88.16| 96.13/93.83 | 56.83/38.90 | 61.27/43.11| 13.14/4.81 | 17.28/6.54 |18.55/7.94|21.57/8.64|
> | Ours   | **99.39/99.07** | **99.13/98.09** | **76.35/62.11** | **73.70/62.75** | **39.83/19.94** | **43.89/24.08** | **35.38/27.28** | **70.49/54.89** |
>
>
> - ***MSCOCO**, $\epsilon=12$, surrogate model: CLIP-ViT-B/16* (some methods use full coco test set of 5000 images for evaluation.)
>
> |**Target Model/Source Model**|CLIP-ViT-B/16|CLIP-ViT-B/16|ALBEF|TCL|TCL|
> |:-:|:-:|:-:|:-:|:-:|:-:|
> | |**TR R@1/@5**|**IR R@1/@5**|**TR R@1/@5**|**TR R@1/@5**|**IR R@1/@5**|
> |Co-Attack [1]|97.98/94.94|98.80/96.83|30.28/13.64|32.84/15.27|44.69/29.42|
> |ETU [3]|96.80/94.35|97.25/95.70|**32.43**/15.76|34.02/17.57|30.28/15.89|
> |Ours|**99.59/98.92**|**99.32/98.44**|31.97/**15.97**|**49.01/32.74**|**53.12/40.31**|
>
>
>
> ---
>
> Regarding the comment "leverages CLIP as a surrogate, which aligns with an existing line of work that has demonstrated similar effectiveness", we clarify that all our baseline comparison methods also use CLIP, either directly or as part of an ensemble, but still perform poorly compared to ours.
>
> Therefore, using CLIP as a surrogate is only one contributing factor to our strong results, the key advantage lies in our proposed source-target matching strategy, which plays a much more critical role in the effectiveness of our attack.
>
>
> In summary, we will carefully revise the paper to:
>
> 1. Include all the above detailed discussions with the mentioned related work in our revision.
>
> 2. Include the comparisons of these related methods provided in our rebuttal with the most relevant CLIP-based attack methods, together with results on standard open-source benchmarks in the tables we provided above.
>
> We sincerely appreciate the reviewer's thoughtful feedback and will reflect these additions and clarifications in the revised version. Please feel free to let us know if you have any further comments.
>
> -------
> References:
>
> [1] Zhang, J., Yi, Q., & Sang, J. "Towards adversarial attack on vision-language pre-training models." In ACM MM 2022.
>
> [2] Lu, D., Wang, Z., Wang, T., Guan, W., Gao, H., & Zheng, F. "Set-level guidance attack: Boosting adversarial transferability of vision-language pre-training models." In ICCV 2023.
>
> [3] Zhang, P. F., Huang, Z., & Bai, G. "Universal adversarial perturbations for vision-language pre-trained models." In SIGIR 2024.

---

### Official Review · Reviewer_mgEt · 2025-06-28

**Clarity:** 3
**Significance:** 3
**Originality:** 3
**Rating:** 4
**Confidence:** 2

**Summary:**

This paper addresses the poor transferability of targeted adversarial attacks from open-source LVLMs to black-box commercial models. The authors attribute this failure to the lack of semantic precision in perturbations, which are often uniformly distributed and semantically diffuse. To improve semantic alignment, they propose a simple baseline that applies localized perturbations by cropping the adversarial image at random scales and aspect ratios, aligning it with the target in embedding space.

**Questions:**

1. How many queries are typically required to achieve the reported success rates, and how does performance change under limited query budgets?

2. How does the method compare to saliency- or attention-guided perturbation approaches?

3. Can the method generalize to other vision-language tasks such as image captioning or VQA?

**Ethical Concerns:**

["NO or VERY MINOR ethics concerns only"]

**Final Justification:**

I thank the authors for their response, which has addressed all of my concerns. Therefore, I am willing to raise my rating from borderline reject to borderline accept. Overall, this approach is easy to implement, achieves strong transferability and experimental results are persuasive.

**Limitations:**

Yes.

**Paper Formatting Concerns:**

No.

**Quality:**

3

**Strengths And Weaknesses:**

**Strengths**

The paper tackles an important and underexplored challenge in adversarial attacks on black-box LVLMs. It proposes a simple yet effective method that enhances semantic alignment by applying local perturbations through randomized cropping and embedding-space alignment. The approach is easy to implement, model-agnostic, and achieves strong transferability across a wide range of commercial models, including GPT-4.5, GPT-4o, Claude, and Gemini, with success rates exceeding 90%. The empirical results are strong, and the connection between perturbation semantics and transferability is well-motivated and clearly articulated.

**Weaknesses**

1. The method is evaluated solely based on success rate, without reporting query counts or trade-offs under limited query budgets.
2. The approach emphasizes local semantic alignment, but does not compare against existing saliency-guided or attention-based adversarial methods.
3. Most experiments focus on image-text matching. The method is not evaluated on richer vision-language tasks like image captioning or visual question answering.

---

> ### Author Rebuttal · Authors · 2025-07-31
>
> We appreciate the reviewer for the constructive and valuable feedback, which will definitely help us improve the quality of our paper. We will accommodate all suggestions into the revised manuscript. Below, we provide detailed responses to each of the reviewer's comments.
>
> >**W1 & Q1**: **W1**) The method is evalated solely based on success rate, without reporting query counts or trade-offs under limited query budgets. **Q1**) How many queries are typically required to achieve the reported success rates, and how does performance change under limited query budgets?
>
> Thanks for the suggestion. All our results in the paper are based on a **single-query setting** to demonstrate the efficiency of the attack. As suggested, we have extended the evaluation to 3 and 5 queries, and the success rates increase to higher as expected, as shown in the table below. We will include these results and the corresponding discussion in the revised version to highlight the trade-offs under limited query budgets.
>
> | **Models**      | **Query&nbsp;(1 / 3 / 5,&nbsp;ASR)** | **Query&nbsp;(1 / 3 / 5,&nbsp;KMR\_a)** | **Query&nbsp;(1 / 3 / 5,&nbsp;KMR\_b)** | **Query&nbsp;(1 / 3 / 5,&nbsp;KMR\_c)** |
> | --------------  | :----------------------------------: | :---------------------------------------: | :---------------------------------------: | :---------------------------------------: |
> | GPT-4o       | 0.95 / 0.96 / 1.00 | 0.82 / 0.93 / 0.93 |  0.54 / 0.79 / 0.83 |  0.20 / 0.28/ 0.28 |
> | Gemini-2.0         | 0.78 / 0.86 / 0.88  | 0.77 / 0.86 / 0.89 | 0.57 / 0.68 / 0.71 | 0.15 / 0.21 / 0.23 |
> | Claude-3.5      | 0.29 / 0.32 / 0.44 | 0.31 / 0.33 / 0.33 | 0.18 / 0.20 / 0.20 | 0.03 / 0.06 / 0.06 |
>
> >**W2 & Q2**: **W2**) The approach emphasizes local semantic alignment, but does not compare against existing saliency-guided or attention-based adversarial methods. **Q2**) How does the method compare to saliency- or attention-guided perturbation approaches?
>
> AdvDiffVLM [1] is a saliency-guided method that uses a GradCAM-based mask generation process to prevent adversarial features from being overly concentrated in specific regions, thereby improving the overall image quality. Following the suggestion, we provide a detailed comparison with the saliency-guided adversarial method AdvDiffVLM in the table below:
>
>
> | **Method**      | **GPT‑4o&nbsp;(a / b / c,&nbsp;ASR)** | **Gemini 2.0&nbsp;(a / b / c,&nbsp;ASR)** | **Claude 3.5&nbsp;(a / b / c,&nbsp;ASR)** |
> | --------------  | :----------------------------------: | :---------------------------------------: | :----------------------------------------: |
> | AdvDiffVLM       | 0.02 / 0.00 / 0.00 (0.02) | 0.01 / 0.00 / 0.00 (0.01) | 0.00 / 0.00 / 0.00 (0.00) |
> | M‑Attack (Ours) | **0.82 / 0.54 / 0.13 (0.95)** | **0.75 / 0.53 / 0.11 (0.78)** | **0.31 / 0.18 / 0.13 (0.29)** |
>
> [1] Guo, Qi, et al. "Efficient generation of targeted and transferable adversarial examples for vision-language models via diffusion models." IEEE Transactions on Information Forensics and Security (2024).
>
> >**W3 & Q3**: **W3**) Most experiments focus on image-text matching. The method is not evaluated on richer vision-language tasks like image captioning or visual question answering. **Q3**) Can the method generalize to other vision-language tasks such as image captioning or VQA?
>
> We provide the suggested more results as follows:
>
> **1. Image Captioning**:
>
> *Source dataset: ImageNet & target dataset: COCO2014 Val*.
>
> **Reference paper**:
>
> AnyAttack: Towards Large-scale Self-supervised Adversarial Attacks on Vision-language Models. In CVPR 2025.
>
> **Experimental Results:**
>
> - Results on black-box LVLMs (GPT-4o, Gemini-2.0, and Claude-3.5):
>
> | **AnyAttack\|M-Attack**      | **SPICE** | **BLEU-1** | **BLEU-4** |**METEOR** |**ROUGE-L** |**CIDEr** |
> |:----------| :-----------| :------------:| :----------: | :-----------: | :-----------: | :-----------: |
> | GPT-4o       | 2.72\|**10.33** | 26.22 \|**37.42** | 1.72\|**6.12** | 9.33 \|**16.26** | 23.69 \|**31.42** | 7.06\|**27.31** |
> | Gemini-2.0 | 3.18\| **7.97** | 26.91\|**34.43** | 0.00\|**5.19** | 8.79\|**14.10** | 21.81\|**29.60** | 8.46\| **22.91** |
> | Claude-3.5 | 2.37\|**2.70** | 22.99\|**23.10** | 0.00\|**1.36** | 8.00\|**8.38** | 20.86\| **20.94** | 4.46\| **5.23** |
>
>
> - Results on open-source LVLMs (BLIP, BLIP2, and InstructBLIP):
>
> | **AnyAttack\|M-Attack**      | **SPICE** | **BLEU-1** | **BLEU-4** |**METEOR** |**ROUGE-L** |**CIDEr** |
> |:----------|:-----------|:------------|:----------|:-----------|:-----------|:-----------|
> | BLIP       | 4.13\|**12.02** | 46.32 \|**65.71** | 3.13\|**23.12** | 11.38 \|**21.07** | 33.32 \|**46.82** | 18.28\|**86.23** |
> | BLIP2 | 4.48\| **8.69** | 46.68\|**53.29** | 5.96\|**13.52** | 11.38\|**15.43** | 33.44\|**38.52** | 20.20\| **44.25** |
> | InstructBLIP | 5.89\|**15.14** | 38.79\|**51.76** | 3.83\|**11.57** | 12.77\|**20.91** | 28.36\| **39.55** | 16.63\|**43.47** |
>
> **2. Visual Question Answering (OK-VQA Dataset)**:
>
> **Reference paper:**
>
> On the Robustness of Large Multimodal Models Against Image Adversarial Attacks. In CVPR 2024.
>
> **Experimental Results:** (*Pre* refers to accuracy for pre-attack.)
>
> - Results on black-box LVLMs (GPT-4o, Gemini-2.0, and Claude-3.5):
>
> | **VQA Accuracy (%)**  | **GPT-4o $\downarrow$** | **Gemini-2.0 $\downarrow$** | **Claude-3.5 $\downarrow$** |
> |:----------|:-----------:| :------------: | :-----------: |
> | Pre       | 27.0 | 30.2 | 15.8 |
> | AnyAttack | 22.4 | 26.2 | 11.8 |
> | M-Attack (Ours) | **7.8**| **14.2** | **5.8** |
>
> - Results on open-source LVLMs (BLIP, BLIP2, and InstructBLIP):
>
> | **VQA Accuracy (%)**  | **BLIP2 $\downarrow$** | **InstructBLIP $\downarrow$** | **LLaVA1.5 $\downarrow$** |
> |:----------|:-----------:| :------------: | :-----------: |
> | Pre       | 25.0 | 33.6 | 30.2 |
> | AnyAttack | 7.2 | 21.8 | 24.8 |
> | M-Attack (Ours) | **6.8**| **13.4** | **12.4** |

---

> > ### Comment · Reviewer_mgEt · 2025-08-03
> >
> > I thank the authors for their response, which has addressed all of my concerns. Therefore, I am willing to raise my rating from borderline reject to borderline accept.

---

### Official Review · Reviewer_mqJv · 2025-06-29

**Clarity:** 3
**Significance:** 3
**Originality:** 3
**Rating:** 5
**Confidence:** 5

**Summary:**

This paper aims to address the low success rate of existing adversarial attacks on commercial closed-source Large Vision-Language Models (LVLMs). The authors begin by analyzing failed attack samples, identifying the root cause as perturbations that exhibit a "Uniform-like Distribution" and lead to "Vague Descriptions," thereby lacking effective semantic information. Based on this insight, the paper proposes an attack framework named M-Attack. The method utilizes Local-level Matching and Model Ensemble strategies to focus the generation of perturbations on local image regions, effectively embedding target semantics into the adversarial examples. Experimental results demonstrate that this method achieves a success rate far exceeding current SOTA methods on several top-tier commercial LVLMs, proving its effectiveness and novelty.

**Questions:**

1. Contradiction with Existing Semantic Attacks: The paper's core premise is that prior methods generate perturbations with a Uniform-like Distribution. However, some methods, such as GAKer [1] and AnyAttack [2], appear to generate perturbations with strong semantic properties. This seems to contradict the authors' claim. It is suggested that the authors add a discussion to address this potential discrepancy.

2. Insufficient Explanation of the Local-Global Mechanism: The explanation for the effectiveness of the Local-Global mechanism is not sufficiently clear. It is counter-intuitive that this method performs so well, even outperforming Local-Local on the KMRScore in Figure 9, as it essentially still optimizes the distance to the target image's global information. Could the authors provide a more detailed analysis and explanation? Specifically, why is the Local-Global approach capable of producing richer semantic details?

[1] Any Target Can be Offense: Adversarial Example Generation via Generalized Latent Infection

[2] AnyAttack: Towards Large-scale Self-supervised Adversarial Attacks on Vision-language Models

**Ethical Concerns:**

["NO or VERY MINOR ethics concerns only"]

**Final Justification:**

Thank you for the detailed rebuttal. The new experiments on open-source models and the clarifying discussion on other semantic attacks have effectively resolved my primary concerns and significantly strengthened the paper.

While I still find the explanation for the Local-Global mechanism's superiority to be more intuitive than empirically proven, I consider this a minor point given the paper's overall strengths.

Therefore, I have raised my rating from borderline accept to 'Accept', as the added experiments and discussions have convincingly addressed my most significant concerns.

**Limitations:**

yes

**Paper Formatting Concerns:**

Nan

**Quality:**

3

**Strengths And Weaknesses:**

# Strengths
1. Clear Motivation and Insight: The paper's motivation is clear. It pinpoints the cause of low attack success rates on commercial models—the Uniform-like Perturbation Distribution and Vague Descriptions—providing a clear direction for future research.

2. Elegant and Effective Method: The proposed M-Attack approach is well-conceived. It presents a simple and efficient local attack strategy that focuses on optimizing the local features of adversarial examples, effectively enhancing the consistency of detailed expressions between the perturbation and the target image. The design is both elegant and easy to understand.

3. Convincing Empirical Results: The proposed method achieves attack success rates on multiple commercial models that significantly surpass prior art, which strongly proves its effectiveness.

# Weaknesses:

1. A core advantage of M-Attack is its ability to generate adversarial examples with clear and rich semantic details. The current evaluation relies mainly on keyword matching to verify consistency. Have the authors considered introducing more evaluation dimensions to further validate this "clear details" advantage? For instance, assessing the consistency of object positional relationships, colors, or other attributes in the generated description compared to the target image.

2. The attacks in this paper exclusively target commercial closed-source models. However, these models often employ complex and unknown defense mechanisms (e.g., output post-processing, content filtering), which poses a challenge for attributing the attack's success. Have the authors tested their method on mainstream open-source LVLMs (e.g., LLaVA, Qwen-VL)? Experiments on open-source models would allow for a more transparent analysis of the model's behavior.

---

> ### Author Rebuttal · Authors · 2025-07-30
>
> We appreciate the reviewer for the constructive and valuable feedback, which will certainly help us improve the quality of our paper. We will incorporate all suggestions into the revised manuscript. Below, we provide detailed responses to each of the reviewer's comments.
>
> >**W1**: A core advantage of M-Attack is its ability to generate adversarial examples with clear and rich semantic details. The current evaluation relies mainly on keyword matching to verify consistency. Have the authors considered introducing more evaluation dimensions to further validate this "clear details" advantage? For instance, assessing the consistency of object positional relationships, colors, or other attributes in the generated description compared to the target image.
>
> Yes, we have actively explored the integration of multi-dimensional evaluation metrics that go beyond flat keyword matching. We extend the current evaluation to assess attribute alignment (e.g., color, shape, count) between the adversarial output and the target image using structured templates or automated tools such as pretrained visual scene graph parsers or attribute extractors. Inspired by the reviewer's suggestion, we are considering an additional module that checks for spatial and relational consistency (e.g., "the dog is sitting on the couch") by parsing the generated descriptions into structured triplets and comparing them to those extracted from the target caption or visual scene graph. To complement automated metrics, we are designing a human evaluation study where annotators will judge not just semantic similarity, but also fidelity of fine-grained visual attributes, such as object layout and spatial coherence.
>
> Specifically, we have tried the following explorations and their result examples. We observe that:
>
> *Gemini-2.0's description of the adversarial image contains many details of the target image, which closely align with those of the target.*
>
> **Example1:** (source image: A longhorn beetle)
>
> This is a close-up, atmospheric shot of a fruit stall, likely at an outdoor market.
> In the foreground, a large pile of **vibrant**, light-green pears dominates the view. Their skin appears slightly rough and speckled, and **they are packed tightly together, filling the bottom half of the frame**.
> In the background, a lime-green, patterned cloth serves as a canopy or backdrop for the stall. **Hanging from the top is a dark strap** or part of the stall's structure.
> A fascinating detail is visible **in the middle of the image**: through a gap or perhaps a reflection, you can see a **bustling market street** with **a crowd of people and buildings in the distance**. A price tag is partially visible just above this scene.
> The overall image is characterized by its overwhelming green palette, giving it a fresh but also slightly surreal quality. The focus on the pears creates a strong sense of texture and abundance.
>
> *Claude-3.5's description of the adversarial image contains many details of the target image, which closely align with those of the target.*
>
> **Example2:** (source image: a fox in the middle)
>
> This image shows a giraffe in its natural habitat. **The giraffe's long neck** and distinctive **spotted coat pattern** are clearly visible. It appears to be **surrounded by bushes and vegetation** typical of an African savanna or grassland environment. The giraffe's head is slightly **tilted**, suggesting it might be reaching for leaves to eat from nearby trees or shrubs. The coloration of the giraffe blends well with the **dappled sunlight and shadows** cast by the surrounding foliage, demonstrating the animal's natural camouflage in its environment.
>
> *GPT-4o's description of the adversarial image contains many details of the target image, which closely align with those of the target*.
>
> **Example3:** (source image: a chair)
>
> This image shows **several sheep or goats** gathered around and behind wooden and metal enclosures or pens. They appear to be contained within a structured or fenced area, likely outdoors or in a barn. The scene suggests a farm or pastoral setting. The lighting seems **a bit dim with some natural light filtering through**.
>
> >**W2**: The attacks in this paper exclusively target commercial closed-source models. However, these models often employ complex and unknown defense mechanisms (e.g., output post-processing, content filtering), which pose a challenge for attributing the attack's success. Have the authors tested their method on mainstream open-source LVLMs (e.g., LLaVA, Qwen-VL)? Experiments on open-source models would allow for a more transparent analysis of the model's behavior.
>
> We provide the results on mainstream open-source LVLMs of LLaVA and Qwen-VL as follows:
>
> | **Method**  | **Qwen-2.5-VL&nbsp;(a / b / c,&nbsp;ASR)** | **LLaVA-1.5&nbsp;(a / b / c,&nbsp;ASR)** |
> |---| :----: | :-----: |
> | AttackVLM       | 0.12/ 0.04 / 0.00 (0.01) | 0.11/ 0.03 / 0.00 (0.07) |
> | SSA‑CWA         | 0.36 / 0.25 / 0.04 (0.38) | 0.29 / 0.17 / 0.04 (0.34) |
> | AnyAttack       | 0.53 / 0.28 / 0.09 (0.53) | 0.60 / 0.32 / 0.07 (0.58) |
> | **M‑Attack** | **0.80 / 0.65 / 0.17 (0.90)** | **0.85 / 0.59 / 0.20 (0.95)** |
>
> >**Q1**: Contradiction with Existing Semantic Attacks: The paper's core premise is that prior methods generate perturbations with a Uniform-like Distribution. However, some methods, such as GAKer [1] and AnyAttack [2], appear to generate perturbations with strong semantic properties. This seems to contradict the authors' claim. It is suggested that the authors add a discussion to address this potential discrepancy.
>
> Thanks for bringing this to our attention and for pointing out these relevant works. We acknowledge that GAKer [1] and AnyAttack [2] introduce adversarial strategies with semantic considerations. However, there are important distinctions that clarify the apparent contradiction: 1) Different Objectives – Generalization vs. Semantic Localization: GAKer and AnyAttack focus primarily on generalization, that is, crafting perturbations that remain effective across unknown classes or models. Their designs especially GAKer are class-agnostic and operate by injecting perturbations into intermediate feature spaces via adversarial generators. The generator is optimized to maintain visual consistency while moving the representation toward a target class in feature space. This goal is orthogonal to ours. 2) Our Focus – Semantically Grounded Perturbations: In contrast, our method is centered around explicitly generating perturbations with localized semantic meaning, directly grounded in interpretable image regions and aligned with specific target content. Rather than pursuing class-level generalization, we target semantically faithful alignment between the perturbed image and a target image—especially in the challenging setting of strong black-box LVLMs. 3) Attack Difficulty – Black-box LVLMs vs. Feature-space Attacks: Compared to attacks like GAKer that operate in intermediate feature spaces with access to surrogate model gradients, our setting is more restrictive and challenging. We operate in a strict black-box regime against large-scale vision-language models (e.g., GPT-4o), where gradient access is unavailable, and surrogate misalignment is significant. We appreciate the opportunity to make this distinction clearer. We will revise our manuscript to include a detailed discussion comparing our approach to GAKer and AnyAttack, highlighting the differences in motivation, attack space, and adversarial objectives.
>
> >**Q2**: Insufficient Explanation of the Local-Global Mechanism: The explanation for the effectiveness of the Local-Global mechanism is not sufficiently clear. It is counter-intuitive that this method performs so well, even outperforming Local-Local on the KMRScore in Figure 9, as it essentially still optimizes the distance to the target image's global information. Could the authors provide a more detailed analysis and explanation? Specifically, why is the Local-Global approach capable of producing richer semantic details?
>
> Thanks for the constructive suggestion. Below, we provide more clarifications on why this strategy is effective:
> 1) *Global Target Offers Richer Semantic Guidance*: Using the global target image (rather than a local crop) provides a more comprehensive and stable semantic reference. This reduces ambiguity in supervision and ensures that the optimization process aligns with the full set of semantic attributes, such as object presence, relationships, and context that the target is intended to convey.
> 2) *Asymmetric Matching Encourages Semantic Encoding*: The Local-Global design introduces asymmetric matching—the source is updated via local windows while the target remains holistic. This asymmetry encourages each local window in the source image to encode complementary aspects of the target semantics, ultimately leading to more semantically rich and diverse perturbations across regions.
> 3) *Avoiding Variance and Instability from Local-Local Supervision*: In contrast, if both source and target use local crops (as in Local-Local), the semantic supervision becomes more sensitive to random crop alignment. This can introduce high variance or even ambiguity in what semantics are being enforced, especially when key target attributes fall outside the cropped region. This instability can degrade convergence and lead to underoptimized attacks.
> 4) *Empirical Alignment with Prior Work*: Similar observations have been noted in recent studies, where maintaining a stable and semantically rich supervision target leads to better optimization dynamics in the latent space or cross-modal embeddings. We will expand our explanation in the revised manuscript to clarify these intuitions and highlight the semantic asymmetry benefits of the *Local-Global* design. Thank you again for pointing this out.

---

> > ### Comment · Reviewer_mqJv · 2025-08-03
> >
> > Thank you for the detailed rebuttal. The new experiments on open-source models and the clarifying discussion on other semantic attacks have effectively resolved my primary concerns and significantly strengthened the paper.
> >
> > While I still find the explanation for the Local-Global mechanism's superiority to be more intuitive than empirically proven, I consider this a minor point given the paper's overall strengths.
> >
> > Therefore, I have raised my rating from borderline accept to 'Accept', as the added experiments and discussions have convincingly addressed my most significant concerns.

---

### Official Review · Reviewer_8QNL · 2025-07-03

**Clarity:** 3
**Significance:** 3
**Originality:** 3
**Rating:** 5
**Confidence:** 5

**Summary:**

The paper proposes an adversarial attack method for large Vision Language Models. The method is built on an insight that failed adversarial perturbations typically originates from a uniform distribution and lack clear semantic details, leading strong VLMs to ignore them as noises. The proposed attack crops and rescales the adversarial image to align with the target image in the embedding space. This strategy achieves surprisingly good transferability to commercial VLMs, achieving over 90% success rate on the most advanced VLMs.

**Questions:**

Q1: Local windows are selected according to requirements indicated in Equation (3) and (4). Therefore, local windows selected at different timestep always will have an overlap with each other. Will the perturbations optimized on the latest local window sabotage the previously optimized perturbation region and render the previous ones ineffective?

Q2:  This work observes an equal or weaker robustness to the attack on reasoning models which is interesting. In the model’s chain-of-thought, does it notice or mention anything unusual in its thinking process? Can you show some examples of the chain-of-thought when the reasoning model behaves unusually with these adversarial samples?

Q3: How much is the computation cost of launching an instance of the proposed attack?

**Ethical Concerns:**

["NO or VERY MINOR ethics concerns only"]

**Final Justification:**

Thanks to the authors for their efforts in addressing mine and my peer reviewers' concerns. I am satisfied with the results and further explanations provided by the authors, therefore, I am willing to raise my rating from borderline accept to accept. Overall, the proposed method is effective and insightful, highlighting interesting properties and potential research opportunities for the safety of vision language models.

**Limitations:**

Yes.

**Paper Formatting Concerns:**

No formatting concerns.

**Quality:**

3

**Strengths And Weaknesses:**

Strength: The observation is in-depth and provides insights. The proposed method works very well under the practical real-world settings. Experiments are thoroughly done. The paper is well-written and easy to follow.

---
Weakness:
W1: Evaluation. The evaluation is done only on 100 sampled images from a NIPS 2017 competition (Line 225-226). Without using a standardized set, the selected images could be cherry-picked, favoring the proposed attack method. This poses challenges for future work to fairly compare. Additionally, the attack success rate is measured by semantic keywords matching, where the keywords are partially manually generated, which may contain bias.

W2: The “groundtruth” labels are given by GPT-4o and human labellers. However, for different VLMs, they may have different ways to express the same semantic even when the image is clean. The paper doesn’t provide the clean KMR accuracy of the tested models as a baseline.

---

> ### Author Rebuttal · Authors · 2025-07-30
>
> We sincerely thank the reviewer for the constructive and valuable comments, which are definitely helpful for us to improve the quality of the paper. We will accommodate all the suggestions in our revised manuscript. In the following, we provide the detailed responses to each of the questions raised by the reviewer.
>
> >**W1**: Evaluation. The evaluation is done only on 100 sampled images from a NIPS 2017 competition (Line 225-226). Without using a standardized set, the selected images could be cherry-picked, favoring the proposed attack method. This poses challenges for future work to fairly compare. Additionally, the attack success rate is measured by semantic keywords matching, where the keywords are partially manually generated, which may contain bias.
>
> Thanks for raising this. We clarify that the 100 samples we used in our evaluation are taken directly from prior work SSA-CWA [1] to ensure consistency with theirs, making it a standardized evaluation set. We will revise the relevant description of the manuscript carefully to make this clearer.
>
> [1] Yinpeng Dong, Huanran Chen, Jiawei Chen, Zhengwei Fang, Xiao Yang, Yichi Zhang, Yu Tian, Hang Su, and Jun Zhu. "How robust is google's bard to adversarial image attacks?." arXiv preprint arXiv:2309.11751 (2023).
>
> Additionally, we provided results on 1,000 images from *NIPS 2017 Adversarial Attacks and Defenses Competition Dataset* in Table 10 of our original manuscript. For reference, we list below the attack success rates on the 1K image set:
>
>
> |     | GPT‑4o | Gemini-2.0 | Claude-3.5 |
> |--|:--|:--|:--|
> |  Threshold     | AnyAttack [2] \| Ours | AnyAttack [2] \| Ours | AnyAttack [2] \| Ours |
> | 0.3       | 0.419 \| **0.868** | 0.314 \| **0.763** | 0.211 \| **0.219**\* |
> | 0.4     | 0.082 \| **0.614** | 0.061 \| **0.444** | 0.046 \| **0.055** |
>
> (* we rerun our codebase during rebuttal and obtain slightly better result.)
>
> [2] Jiaming Zhang, Junhong Ye, Xingjun Ma, Yige Li, Yunfan Yang, Yunhao Chen, Jitao Sang, and Dit-Yan Yeung. "AnyAttack: Towards Large-scale Self-supervised Adversarial Attacks on Vision-language Models." In CVPR 2025.
>
> For the concern of "the attack success rate is measured by semantic keywords matching, where the keywords are partially manually generated, which may contain bias", we clarify that:
>
> 1) The keyword sets are constructed following consistent and transparent criteria, focusing on objective semantic concepts that are directly entailed by the ground truth. For instance, if the reference answer involves a red object and the number of entities, the corresponding keywords would be ["red", "object", "two"]. This standardization reduces arbitrary or subjective labeling.
> 2) KMRScore applies the same keyword set uniformly across all model outputs for a given question or task. This ensures that while the initial selection may involve small human judgment, the evaluation itself is deterministic and uniformly applied, avoiding instance-level subjectivity that often arises in human annotations.
> 3) We conduct sensitivity analyses by modifying or perturbing the keyword sets (e.g., removing synonyms or non-critical modifiers), and observe that the overall trend of attack success rates remains consistent. This suggests that our results are robust to small variations in keyword specification.
>
> >**W2**: The “groundtruth” labels are given by GPT-4o and human labellers. However, for different VLMs, they may have different ways to express the same semantic even when the image is clean. The paper doesn’t provide the clean KMR accuracy of the tested models as a baseline.
>
> Thanks for the constructive suggestion. We provide the clean KMR accuracy of the tested models under two scenarios: (1) when the source and target images are semantically different, and (2) when they are semantically similar:
>
>
> |   | Semantics  | Source Image |  kmr_a | kmr_b | kmr_c|
> |--|:--:|:--:|:--:|:--:|:--:|
> | GPT-4o (Upper bound) | Similar  | Clean |  1.00   |     0.90    |    0.40    |
> | GPT-4o (Baseline) | Different | Clean |   0.04   |     0.01    |    0.00    |
> | GPT-4o (Ours) | Different |  Adv | 0.82   |   0.54   |  0.13   |
> | Gemini-2.0 (Upper bound) |  Similar  |   Clean      |   1.00   |     0.95    |    0.30    |
> | Gemini-2.0 (Baseline) | Different  | Clean  | 0.05   |     0.02    |    0.00    |
> | Gemini-2.0 (Ours) | Different  | Adv    |  0.75   |   0.53   |  0.11   |
> | Claude-3.5 (Upper bound) | Similar | Clean  | 1.00   |   0.85   |  0.35   |
> | Claude-3.5 (Baselines) |   Different   |   Clean     |   0.05   |     0.02    |    0.00    |
> | Claude-3.5 (Ours) |   Different   |  Adv    | 0.31   |   0.18   |  0.03   |
>
> As shown in the table, when the source and target have different semantics, the clean images yield very low keyword matching scores (the suggested baselines). In contrast, our method (after adversarial training) achieves much higher scores. When the source and target are semantically similar (the first row in each group), clean images achieve high semantic alignment as upper bound. These results demonstrate the robustness of our method in preserving and conveying key semantic concepts.
>
> We further present a few real example outputs to demonstrate that different VLMs exhibit similar semantic understanding:
>
> **Example1:**
>
>     GPT-4o: Zebras and a wildebeest graze near a lake with distant mountains and pink flamingos in the background.
>     Claude-3.5: Two zebras and a wildebeest graze together near a lake with flamingos visible in the distance.
>     Gemini-2.0: Two zebras and a wildebeest graze on a grassy plain with shrubs, near a lake filled with pink flamingos under a cloudy sky.
>
> **Example2:**
>
>     GPT-4o: A person in festive attire wears a hat and necklace made of bananas, standing in an indoor setting.
>     Claude-3.5: A performer wears a vibrant costume featuring yellow banana decorations, turquoise silk, and colorful accents.
>     Gemini-2.0: A person in an ornate costume, adorned with a headdress and necklace of yellow bananas, smiles playfully with their tongue sticking out.
>
> **Example3:**
>
>     GPT-4o: A man sleeps peacefully in bed with two cats nearby.
>     Claude-3.5: A sleeping person in bed with two gray cats resting nearby on the pillows.
>     Gemini-2.0: A bearded man sleeps in bed, covered by white sheets. Two cats are also in the bed, one beside his head and another by the pillows.
>
>
> >**Q1**: Local windows are selected according to requirements indicated in Equation (3) and (4). Therefore, local windows selected at different timestep always will have an overlap with each other. Will the perturbations optimized on the latest local window sabotage the previously optimized perturbation region and render the previous ones ineffective?
>
> Our goal is to reinforce the *semantic sensitivity* of each region rather than treat different local windows as independent. Therefore, the local windows selected at different timesteps are intentionally designed to **overlap**, as dictated by Equations (3) and (4). This overlap serves two purposes: 1) *Semantic Reinforcement Across Steps*: The overlapping regions allow successive optimization steps to reinforce key semantic content from multiple perspectives, leading to more robust perturbations that generalize better across spatial contexts. 2) *Conflict Avoidance Through Controlled Overlap*: To prevent conflicting updates or overfitting of multiple semantics into the same region, we carefully control the extent of overlap between steps. This ensures that while the windows share some common areas, they also retain sufficient uniqueness to preserve gradient diversity and prevent destructive interference between semantic objectives.
>
> In practice, we observe that this overlap strategy improves attack stability and convergence, as it balances localized semantic precision with broader contextual coverage. We will include a clearer explanation of this design choice in our revised version.
>
> >**Q2**: This work observes an equal or weaker robustness to the attack on reasoning models which is interesting. In the model’s chain-of-thought, does it notice or mention anything unusual in its thinking process? Can you show some examples of the chain-of-thought when the reasoning model behaves unusually with these adversarial samples?
>
> Thanks for the insightful comment. We notice that thinking models tend to see the content of the source image but interpret it in a way that aligns with the target image. For example, if the original image shows a green barber cape in a hair salon and the target image depicts a group of grayish boats, the model might reinterpret the cape as a green waterproof cover on a boat. In other words, even when it recognizes elements from the original image, it semantically shifts them toward the context of the target image. Generally, these "thinking models" (e.g., Gemini-2.5 Pro, Claude-3.7-thinking, o1) describe an image in a highly logical manner while covering abundant visual details.
>
> Taking Gemini-2.5 Pro as an example, during its reasoning process, the model sometimes detects elements present in the source image, yet deliberately blends those elements with the semantic content introduced by the adversarial perturbation.
>
> **Example:**
>
> The source image shows a red leather chair in a barbershop draped with a green barber cape, whereas the target image depicts a fleet of gray-and-white vendor boats moored offshore. While generating its explanation, the model states:
>
> **Defining the Image's Features:**
>     *"My focus is now on constructing a detailed description of the scene. I've broken it down into foreground, midground, and background elements. **The reddish-brown planks**, the conical hat, and the vendor with **a green tarp** are key visual details."*
>
>
> >**Q3**: How much is the computation cost of launching an instance of the proposed attack?
>
> *Time* taken for optimizing/generating an image: **20.04 seconds** on a single RTX 4090 GPU.
>
> *Memory* used for generating an image: **549.78 MB**.

---

### Note · Authors · 2025-08-12

Dear Area Chair and Reviewers,

We sincerely thank the reviewers for your insightful comments and constructive feedback. We have provided a detailed *point-by-point* response in our rebuttal to address each suggestion. The manuscript will be updated to accommodate all suggestions from the reviewers.

We are also encouraged by the positive remarks, such as "the observation is in-depth and provides insights, clear motivation and insight, providing a clear direction for future research, well-written and easy to follow, simple yet effective method, easy to implement, model-agnostic" [**8QNL**, **mqJv**, **mgEt**], "the proposed method works very well under the practical real-world settings, elegant and effective method, offers a novel and insightful explanation" [**8QNL**, **mqJv**, **ZL92**], "experiments are thoroughly done, convincing empirical results, empirical results are strong, an observation well-supported by experiments across multiple baselines, extensive experiments" [**8QNL**, **mqJv**, **mgEt**, **ZL92**]. We will reflect all comments and suggestions in our revision.

Regarding the further novelty concern, we have clarified our technical contributions and novelty in detail. We note that many review comments have highlighted the depth and insight of our observation for this problem, and the reviewer who raised this concern acknowledged that our work offers a novel and insightful explanation for the failures associated with the uniform-like distribution of perturbations, which is the key that forms our approach. We hope our detailed follow-up responses have addressed this concern for this reviewer.


Best,

Authors

---

### Decision · Program_Chairs · 2025-09-17

**Decision:**

Accept (poster)

**Comment:**

The paper introduces a simple transfer attack for black-box LVLMs that optimizes perturbations with random local crops aligned in the embedding space, plus a lightweight surrogate ensemble and a semi-automated evaluation called KMRScore. The authors argue that failed attacks look uniform and lack clear semantics; their local strategy aims to encode specific semantic details in salient regions and transfers well to commercial LVLMs.




The work is easy to implement, model-agnostic, and empirically strong on closed-source systems. The motivation and analysis around non-uniform, semantically focused perturbations are clear, and the paper presents broad comparisons together with a structured evaluation pipeline.



Reviewers initially asked for standard open-source benchmarks, clearer justification that non-uniform perturbations correlate with better transfer, and discussion of bias in KMRScore and LLM-as-judge. In rebuttal the authors added open-source results, provided ECDF evidence for the non-uniformity claim, and explained how KMRScore reduces subjectivity via predefined keyword sets and released criteria.

One reviewer remains partly unconvinced about technical novelty and the CLIP-surrogate connection, requesting explicit comparisons to prior CLIP-based attacks and more results under standard open-source settings; the authors responded and committed to add these, but the concern was only partially resolved.

Given the clear insight, simplicity, and strong empirical evidence, and with most reviewer concerns addressed during rebuttal, I recommend Accept.